# AVOIDDS: Aircraft Vision-based Intruder Detection Dataset and Simulator

**Elysia Q. Smyers**
Department of Computer Science
Stanford University
elysia@cs.stanford.edu

**Sydney M. Katz**
Department of Aeronautics and Astronautics
Stanford University
smkatz@stanford.edu

**Anthony L. Corso**
Department of Aeronautics and Astronautics
Stanford University
acorso@stanford.edu

**Mykel J. Kochenderfer**
Department of Aeronautics and Astronautics
Stanford University
mykel@stanford.edu

## Abstract

Designing robust machine learning systems remains an open problem, and there is a need for benchmark problems that cover both environmental changes and evaluation on a downstream task. In this work, we introduce AVOIDDS, a realistic object detection benchmark for the vision-based aircraft detect-and-avoid problem. We provide a labeled dataset consisting of 72,000 photorealistic images of intruder aircraft with various lighting conditions, weather conditions, relative geometries, and geographic locations. We also provide an interface that evaluates trained models on slices of this dataset to identify changes in performance with respect to changing environmental conditions. Finally, we implement a fully-integrated, closed-loop simulator of the vision-based detect-and-avoid problem to evaluate trained models with respect to the downstream collision avoidance task. This benchmark will enable further research in the design of robust machine learning systems for use in safety-critical applications. The AVOIDDS dataset and code are publicly available at https://purl.stanford.edu/hj293cv5980 and https://github.com/sisl/VisionBasedAircraftDAA, respectively.

## 1 Introduction

The use of machine learning in high stakes applications will require the design of robust systems that perform well in a wide range of environmental conditions [1]–[3]. For example, a learning-based perception system designed for use in aviation will need to handle changing environmental conditions such as weather and time of day. In addition, these systems are often components of an autonomy stack, and their performance should be evaluated both in isolation and with respect to the downstream task of the system in which they operate [4], [5]. For the aviation example, perception models should not only be evaluated based on isolated metrics on a test set such as mean average precision but also on downstream tasks such as collision avoidance. Benchmark systems that allow for this comprehensive evaluation are needed to build robust machine learning systems.

A number of recent benchmark datasets such as WILDS [3] and MetaShift [6] contain distribution shifts due to environmental changes; however, these datasets only allow for evaluation on a test set and generally do not provide evaluation capabilities for downstream decision-making tasks. While recent work has highlighted the importance of task-specific design and evaluation, there is a lack of standardized and accessible benchmarks in this area [4], [5], [7], [8]. To provide a benchmark environment that covers both environmental changes and task-specific evaluation, we

37th Conference on Neural Information Processing Systems (NeurIPS 2023) Track on Datasets and Benchmarks.

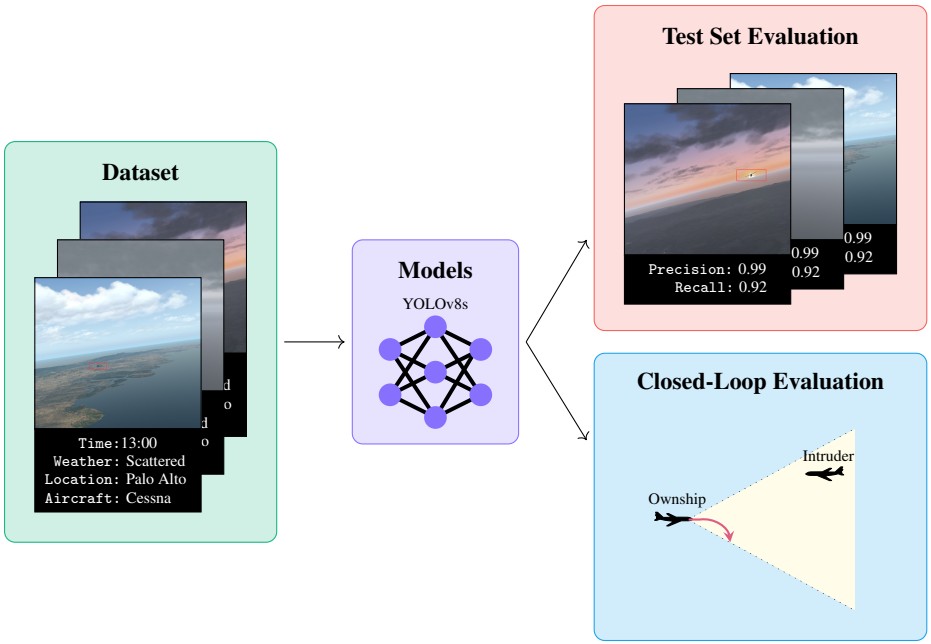

Figure 1: AVOIDDS benchmark overview.

introduce AVOIDDS (Aircraft Vision-based Intruder Detection Dataset and Simulator). AVOIDDS is a benchmark dataset and evaluation environment for the problem of vision-based aircraft detect-and-avoid (DAA). For this task, onboard aircraft systems must detect nearby aircraft and determine proper maneuvers to avoid colliding with them. AVOIDDS specifically focuses on vision-based models, which are trained to detect intruding aircraft from images taken by a mounted camera [9]–[13]. These models will be especially critical in the development of unmanned aircraft and future air mobility concepts.

AVOIDDS allows for training and testing of vision-based aircraft detection models. The benchmark includes three main components (see fig. 1):

- **Data:** We provide a dataset containing 72,000 airspace images for training detection models along with data generation code for customizable creation of new datasets. The dataset is annotated with metadata that describes the environmental conditions under which each image was captured.
- **Baseline models:** We provide baseline YOLOv8 aircraft detection models trained on the AVOIDDS dataset.
- **Evaluation capabilities:** We provide capabilities for test set evaluation and an aircraft encounter simulator to evaluate the closed-loop performance of trained detection models on the downstream task of collision avoidance.

AVOIDDS serves as a benchmark for vision-based aircraft DAA and will enable further research on the robust design and evaluation of learning-based perception systems. By defining a specific downstream task and collecting annotated data across a variety of conditions, we enable further research into solving safety-critical problems.

## 2 Related Work

The AVOIDDS benchmark is related to previous work involving object detection benchmarks, task-specific evaluation, datasets with distribution shifts, and vision-based aircraft detect-and-avoid.

**Object detection benchmarks** Early object detection datasets include Pascal VOC [14] and COCO [15], which contain images of common objects with their corresponding labels. These datasets have

served as standard evaluation benchmarks for new object detection algorithms throughout the last decade [16]–[23]. While these benchmarks cover a wide range of common objects, they do not capture the performance of object detection algorithms on task-specific domains. Li *et al.* [24] created the ODinW (Object Detection in the Wild) dataset, which contains 13 task-specific datasets that were used to check zero-shot performance of language-image models. Ciaglia *et al.* [25] provide a more extensive version of this dataset called RF100 (Roboflow-100) that consists of 100 task-specific object detection datasets from domains such as microscopic images and video games. However, ODinW and RF100 do not provide interfaces to assess task-specific performance metrics on each dataset. In this work, we provide both a task-specific dataset for vision-based aircraft collision avoidance and a simulator to evaluate performance on the downstream collision avoidance task.

**Task-specific evaluation**  Object detection models are often evaluated in isolation using metrics such as precision, recall, and mean average precision (mAP) [21]–[23]. However, these models are often components of a system for a high-level decision-making task. For the vision-based detect-and-avoid example, the object detection model is trained to detect intruding aircraft such that the overall system can recommend safe collision avoidance maneuvers. Furthermore, the best model according to traditional metrics does not always result in the best performance on downstream tasks [4], [5], [7], [8]. For this reason, it is important to incorporate task-aware metrics into the design and evaluation of machine learning models. Philion *et al.* [4] develop planner-centric metrics for a machine-learning based object detection system used in autonomous driving and show their benefit over traditional metrics. They evaluate the metrics using driving trajectories from the nuScenes dataset [26]. The nuScenes dataset provides trajectories for the autonomous driving task, but the trajectories are prerecorded and therefore do not allow for a proper simulation to assess the performance of the fully-integrated, closed-loop system [26]. The CARLA simulator enables closed-loop simulation of driving trajectories and scenarios to evaluate various models for perception and control [27]–[30]. In the aviation domain, AirSim allows for the closed-loop simulation of unmanned aerial vehicles, such as small drones [31]–[34]. Corso *et al.* [5] design safer perception systems by quantifying the effect of perception errors on the performance of a downstream task and apply their methods to increase the safety of a vision-based detect-and-avoid system. This work expands on this application to provide an accessible benchmark for task-specific evaluation.

**Datasets with distribution shifts**  Previous work has shown that machine learning models often drop in performance when their test distribution differs from their training distribution [1]–[3]. This dropoff is especially apparent if the model relies on spurious correlations in the training data [35]. For this reason, researchers have created benchmark datasets with distribution shifts. Early datasets containing distribution shifts focused on local perturbations such as rotation and image noise. For example, some works created distribution shifts by rotating images in standard benchmark datasets such as MNIST and CIFAR-10 [36], [37]. Hendrycks *et al.* [38] created the ImageNet-C dataset by applying 75 common corruptions to the ImageNet dataset and noted a drop in performance of state-of-the-art models when evaluated on the corrupted images. Datasets such as WILDS [3] and MetaShift [6] go beyond image tranformations and gather data containing more general distribution shifts that machine learning models may encounter when deployed in the wild. The WILDS dataset, for example, contains 10 datasets with distribution shifts involving domain generalization and subpopulation shift [3]. However, the WILDS dataset does not have clear annotations of the semantic concepts that change in each distribution shift. Datasets such as NICO [39] and MetaShift [6] provide metadata describing the semantic concepts present in images of common objects. This metadata enables the evaluation of model performance across groupings of the dataset with similar concepts. Inspired by the metadata used in NICO and MetaShift, AVOIDDS provides annotations of the environmental conditions such as time of day, weather, and geography for each aircraft image. While previous distribution shift benchmarks evaluate changes in model performance using standard test set metrics, AVOIDDS also allows for evaluation of performance with respect to the downstream task of the broader system in which the model operates.

**Vision-based aircraft detect-and-avoid**  Aircraft detect-and-avoid systems rely on sensor information to detect and track intruding aircraft so that they may issue proper collision avoidance advisories. Traditional sensors used for surveillance and tracking include ADS-B, onboard radar, and transponders [40], [41]; however, automated aircraft collision avoidance systems will require additional sensors both for redundancy and to replace the visual acquisition typically performed by the pilot. For this reason, vision-based traffic detection systems have been proposed, in which intruding aircraft

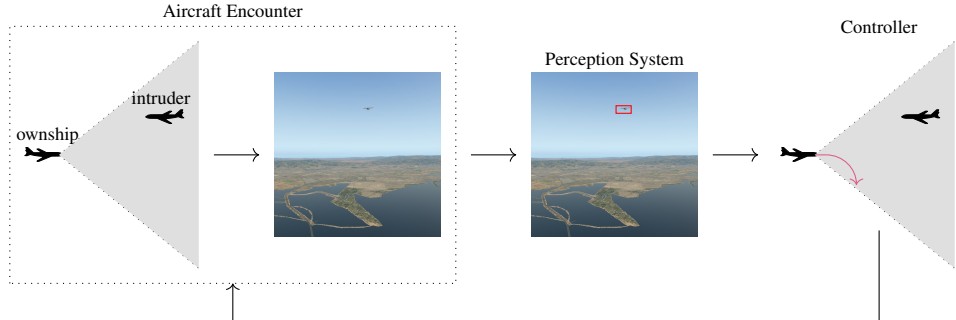

Figure 2: Vision-based detect-and-avoid system overview.

are detected from a camera sensor mounted on the aircraft [9]–[13], [42], [43]. Early work on vision-based aircraft detection used traditional model-based computer vision techniques [42], [43]; however, these techniques were unable to detect aircraft below the horizon in front of a background with ground clutter. Deep learning-based approaches address this limitation and are well-suited for the aircraft detection task [9]–[13]. However, since this application is safety-critical, it is important to evaluate the safety and robustness of these systems with respect to changes in the environment using realistic datasets and simulators [44]. Aerial object tracking datasets provide video sequences of aerial imagery [45]–[47]. The simulator in the AVOIDDS benchmarks produces video sequences similar to these existing tracking datasets while also allowing for the simulation of new scenarios. Of the vision-based traffic detection benchmarks and datasets that are publicly available, AVOIDDS represents the only benchmark with both an extensive dataset covering a range of environmental conditions annotated with metadata and a simulator for evaluation on the downstream collision avoidance task.

## 3   AVOIDDS Benchmark

An aircraft detect-and-avoid system is responsible for sensing nearby aircraft and determining the necessary maneuvers to avoid collision. Figure 2 provides an overview of this process for a vision-based system. At each time step, the equipped aircraft (referred to as the ownship) captures an image of its surroundings using a mounted camera. This image is then passed through a perception system that detects surrounding aircraft (referred to as intruders) and produces an estimate of their state. This state estimate is then passed to a controller, which selects a collision avoidance maneuver. While extensive previous work has studied aircraft collision avoidance controllers that rely on state information [40], [41], [48], [49], the image-based aircraft detection problem remains an open area of research. Therefore, the AVOIDDS benchmark focuses mainly on the detection component of the detect-and-avoid system but also provides a framework for evaluating this component within the context of the fully-integrated, closed-loop system shown in fig. 2.

Figure 1 outlines the three main components of the AVOIDDS benchmark. The first component provides capabilities for the controllable generation of labeled airspace images under a range of environmental conditions using a photo-realistic flight simulator. We used these capabilities to produce a large, public dataset of airspace images with intruder aircraft at various locations in the frame. Using this dataset, we trained baseline YOLOv8 models to detect intruder aircraft in individual frames passed to the model. The final component involves two evaluation systems: one that outputs traditional metrics such as precision and recall evaluated on a test set of individual image inputs and an aircraft encounter simulator that outputs task-specific metrics. With both of these evaluation systems, we can evaluate model performance on sample inputs and on the downstream task that the model is intended to serve.

### 3.1   Data Generation

A challenge for the use of computer vision in aviation is that the variability of environmental variables may lead to lower performance, posing large risks. During flight, aircraft can experience changes in terrain, lighting, and weather that will impact the performance of a vision-based model. To capture this variability, we provide an accessible interface for generating customized image datasets with

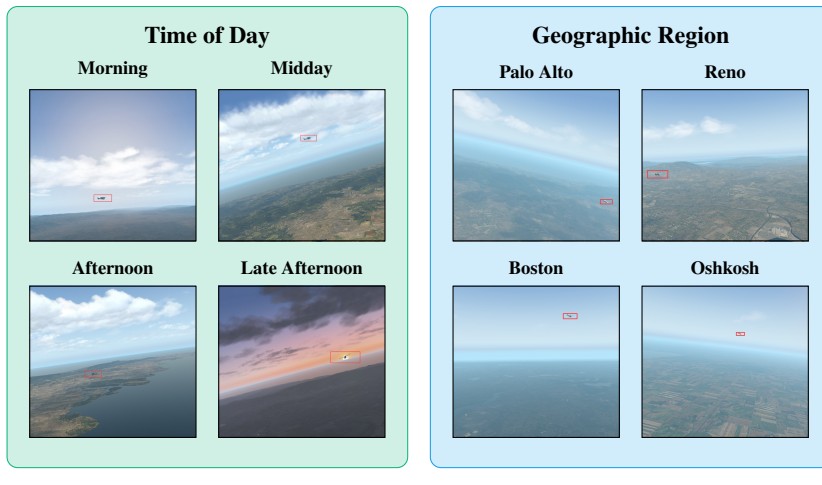

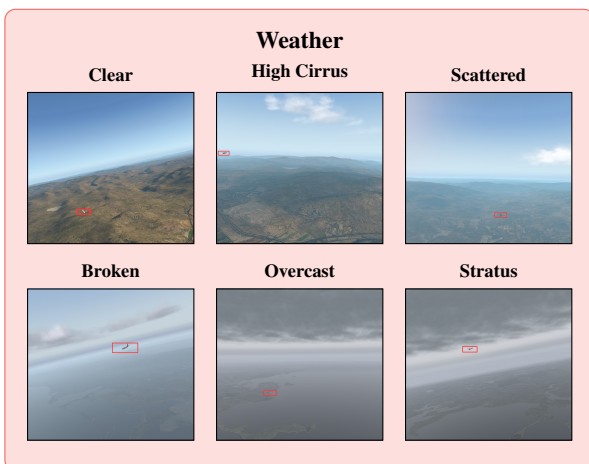

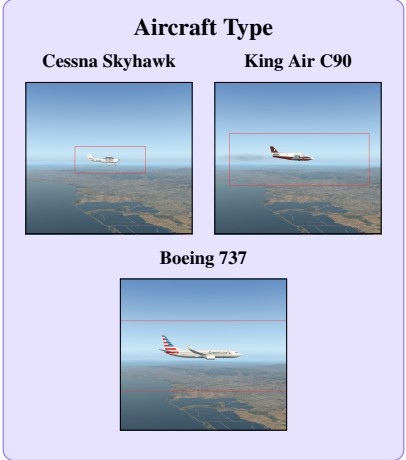

Figure 3: AVOIDDS dataset overview.

a wide range of conditions, including intruder aircraft type, weather, geographic region, relative geometry, and time of day. We use X-Plane 11 [1], a photo-realistic flight simulator, for data generation, allowing for programmatic control of the environmental conditions and intruder location for the rendered images. X-Plane 11 also outputs ground truth position and camera data, which allows us to automatically generate bounding box labels, removing the need for manual labeling. This publicly-available commercial flight simulator has been used in prior work to generate data for an aircraft taxi scenario [50].

Using our accessible interface, we generated the AVOIDDS dataset to adequately cover the aforementioned environmental variations. The AVOIDDS dataset is a collection of 72,000 images and labels from the ownship's point of view of encounters with intruder aircraft in the airspace. Each image is randomized across a range of intruder aircraft types, locations, weather conditions, and times of day with the intruder aircraft located uniformly within the ownship's field of view. Capturing a wide variety of conditions allows us to train an associated model that accounts for these same variations in the environment, which is essential in high-stakes situations such as aircraft collision avoidance.

The 72,000 images generated for this dataset are distributed equally among 6 weather types, 3 aircraft types, and 4 regions. Figure 3 summarizes the various conditions present in the AVOIDDS dataset. The time of day for each sample was randomized between 08:00 and 17:00 on January 1st in each respective location's local time. This range includes an adequate spread of times that represents nominal lighting conditions with a small portion of samples captured around dusk or dawn. The encounter location was sampled uniformly within the surrounding region of the following four airports: Palo Alto (PAO), Reno-Tahoe (RNO), Boston Logan (BOS), and Whitman Regional (OSH).

---

[1]https://www.x-plane.com (Python interface at https://github.com/nasa/XPlaneConnect)

We selected these regions to create variability in the scenery determined by the geography of each region. The range between the ownship and intruder aircraft was sampled from a gamma distribution with a slight skew toward closer ranges. To account for the larger size of the Boeing 737-800 aircraft relative to the smaller aircraft, the gamma distribution was skewed toward slightly larger ranges for Boeing 737-800 images. The vertical and horizontal position of the intruder was sampled uniformly within the ownship field of view.

The AVOIDDS dataset serves as an independent, accessible dataset that can be used for training vision-based object detection models for aircraft collision avoidance without having to interface directly with the X-Plane 11 flight simulator. The dataset abides by the YOLO format [23] with subdirectories for images and labels for both the training and validation set. We also include metadata for each image, containing positional information about the ownship and intruder as well as details about the environment (e.g. weather, time of day, region). The ability to filter the dataset using this metadata enables easy slicing of the data for model training and evaluation purposes. For instance, we can choose to evaluate a model on images from a particular location or time of day or even images with the intruder in specific orientations relative to the ownship. This capability allows for evaluation of the model not only on nominal conditions in the airspace but also on conditions that might result in unpredictable model performance.

### 3.2   Baseline Models

We trained a baseline YOLOv8 object detection model on the AVOIDDS dataset for $100$ epochs with default hyperparameters. We used the YOLOv8s architecture, which includes $11.2$ million trainable parameters. The training took $73\,\mathrm{h}$ on an NVIDIA GeForce GTX 1070 Ti. We also trained an alternative model for comparison on a subset of the AVOIDDS dataset, only including samples in nominal conditions: minimal cloud cover (clear, high cirrus, or scattered clouds) between 08:00 and 15:00 in Palo Alto. The alternative model uses the same architecture and required $6.5\,\mathrm{h}$ for $100$ epochs with default hyperparameters on $6944$ samples.

### 3.3   Evaluation

As shown in fig. 1, AVOIDDS provide two methods for evaluating trained models: evaluation on a test set and evaluation with respect to the downstream task. For the former, we evaluate the performance of detection models on a test set using standard metrics such as precision, recall, and mean average precision (mAP). Evaluation with respect to the downstream task, on the other hand, allows us to see how these standard metrics translate to performance in simulated aircraft encounters using X-Plane 11. This method of evaluation uses task-specific metrics to measure the model's performance.

**Simulator Overview**   The AVOIDDS closed-loop simulator allows us to test vision-based aircraft detection models on the downstream task they were meant to serve: navigating encounters with other aircraft in the airspace. To evaluate closed-loop performance, we provide the three components of the problem shown in fig. 2. We define an encounter model from which the simulator can sample sets of pairwise encounters between an ownship and intruder. We also define a perception system that relies on our trained detection models to predict the intruder state. Its predictions are then passed to a controller produced in previous work [51] that determines the best course of action based on the intruder state. By simulating a full set of encounters and determining the number of encounters that resulted in a near mid-air collision (NMAC), we can evaluate the performance of the detection model with respect to the full closed-loop system. The encounter model only provides positional information and velocities for the two aircraft, allowing for custom simulation of the encounters with different regions, times of day, weather, and intruder aircraft types.

**Encounter Model**   Monte Carlo analysis on airspace encounter models has been used extensively to assess the safety of aircraft collision avoidance systems [52], [53]. Encounter models are probabilistic representations of typical aircraft behavior during a close encounter with another aircraft. To analyze the safety of a particular collision avoidance system, we can simulate the system on a set of encounters and analyze the resulting trajectories. We provide a model that generates pairwise encounters in which the ownship and intruder follow straight line trajectories with various relative geometries. We sample encounters by first sampling features such as aircraft speeds, miss distances, and relative headings. We then use these features to generate trajectories for the ownship and intruder aircraft.

Appendix A provides additional details on this model. We provide this straight-line model as a baseline to demonstrate the evaluation capabilities that AVOIDDS enables, and we define a general interface between the encounter model and simulator such that more complex encounter models can easily be incorporated. For example, recent work in airspace modeling has resulted in a number of publicly available data-driven statistical encounter models that capture the full set of variations in aircraft behavior [54]–[57].

**Perception System** The perception system involves two steps: detecting the intruder in view and interrogating the intruder for its location relative to the ownship. The image observations for the perception system are obtained by positioning the aircraft in X-Plane 11 according to the current state. Once the aircraft are positioned, the perception system uses the vision-based detection model to detect intruding aircraft. If detected, the state of the intruder is then passed to the controller.

**Controller** The controller takes in the intruder state estimated by the perception system and selects an appropriate collision avoidance advisory. Example advisories include "Clear of Conflict" (COC) if no change of course is required and vertical advisories to climb or descend at different rates. We provide an interface to use the control policy defined in the VerticalCAS repository created by Julian *et al.* [51]. VerticalCAS contains an open-source collision avoidance logic loosely inspired by the vertical logic used in a family of collision avoidance systems called ACAS X, which model the collision avoidance problem as a Markov decision process (MDP) [40], [41], [48], [49]. The MDP is solved offline using dynamic programming, which results in a collision avoidance policy that balances between safety and efficiency [58]. The collision avoidance logic is stored as a numeric lookup table, and the ownship looks up its current state during flight based on the relative geometry of the intruder to determine optimal collision avoidance advisories. While VerticalCAS is a strong baseline inspired by real-world systems such as ACAS X, it is a notional example designed for research purposes, and it has not been put through the same rigorous testing and validation as ACAS X.

**Simulator Metrics** We provide capabilities to retrieve task-specific metrics from the simulation results related to safety and efficiency. We assess safety by determining the number of encounters that resulted in an NMAC, defined as a simultaneous loss of separation to less than $500\,\text{ft}$ horizontally and $100\,\text{ft}$ vertically. In this application, we want to minimize the number of NMACs while issuing as few alerts (advisories other than COC) as possible. To evaluate this balance, we also compute the alert frequency, defined as the fraction of time steps in which the system issues an alert.

## 4 Experiments

Using our evaluation capabilities, we evaluate the trained models outlined in section 3.2 on a test set and with respect to the aircraft collision avoidance task.

### 4.1 Test Set Evaluation

We can evaluate the precision, recall, and mean average precision (mAP) of the baseline models on different slices of the dataset. Figure 4 provides a summary, and table 4 in appendix C contains the full results. The baseline model performs strongly, showing an mAP of $0.866$ overall and a precision of over $0.990$ across all categories. In comparison, we see in fig. 4 that the alternative model performs worse overall than the baseline, producing a lower mAP in every category; however, it still makes detections in conditions that were not present during training, indicating some degree of generalizability. The overall lower performance of the alternative model reinforces the need for models to be trained on comprehensive datasets, particularly when used for high-stakes tasks.

There are common patterns between the baseline and alternative models that demonstrate the potential for performance differences within categories, even when trained on a comprehensive dataset. For instance, both models perform worse as the distance between the aircraft increases. In the same way, Boeing 737-800 intruders were easiest to detect by both models even though training was spread evenly between the three aircraft. This result is likely due to the relative size of Boeing 737-800 aircraft compared to the Cessna Skyhawk and King Air C90 and highlights that performance may vary even when model training accounts for variation in conditions. One pattern shown by our test set evaluation environment that we did not intuit was both models performing worse on clear weather conditions than most of the other cloud variations. There are a few potential explanations for

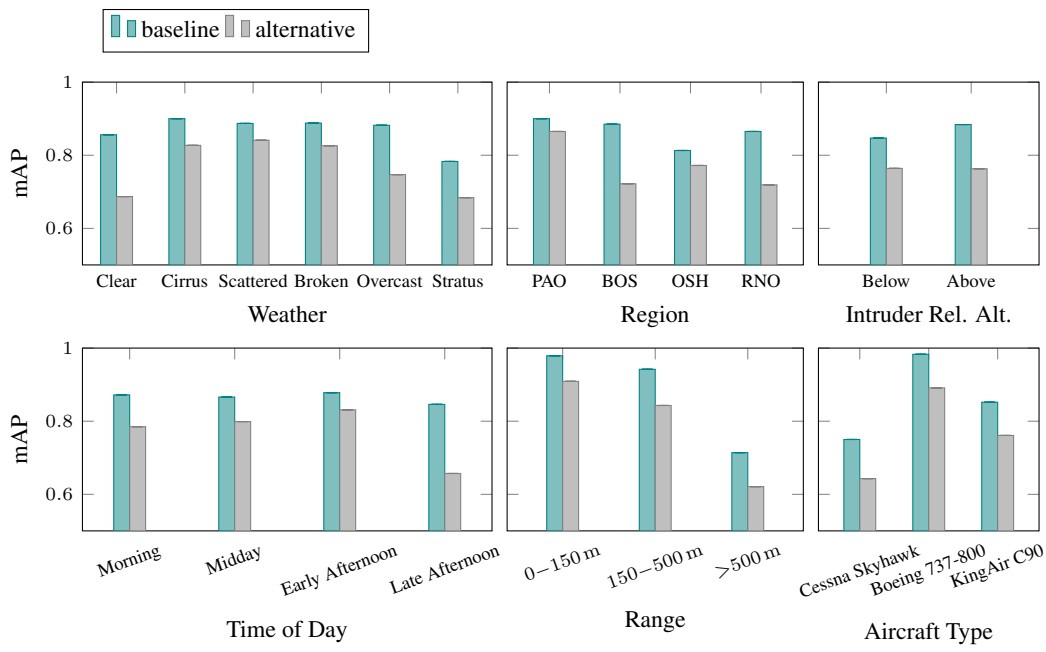

Figure 4: Mean average precision (mAP) of the baseline and alternative models.

this behavior, including that the intruder tends to camouflage with the terrain more when the air is completely clear or that clouds create a more pale background against which intruder detection is made easier by a higher contrast in colors.

In addition to the similarities between the models, we see some differences that further support the argument for comprehensive training. For example, while the models result in a similar mAP on images from the Palo Alto region, there is a significant drop in mAP between the baseline and alternative models for the regions that were not used in the alternative model training. Similarly, the performance of the alternative model decreases significantly for late afternoon samples, in contrast to the baseline model performance. Some late afternoon samples show darker conditions (around dusk), so a drop in performance is not unexpected. However, we see that training the baseline on late afternoon images resulted in more consistent and higher performance on that category. Likewise, training on all locations enabled the baseline model to achieve higher performance than the alternative model. These results demonstrate the importance of comprehensive training datasets.

## 4.2 Downstream Task Evaluation

We evaluated the AVOIDDS baseline model using the simulator on 8640 encounters sampled from the model described in section 3.3 and appendix A. These encounters were equally distributed among each cloud type, region, time of day, and aircraft type shown in fig. 4. For each combination of conditions (of which there are 288), 30 encounters were randomly generated using the encounter model and used to evaluate the detection models. Our aim was to produce analogous results to the test set evaluation, creating the same variation in conditions and at least as many instances to evaluate (8640 encounters and 7200 validation images).

Figure 5 summarizes the safety results, and table 5 in appendix C contains the full safety and efficiency results. Overall, the results highlight the importance of both test set and downstream task evaluation for complex models. We see some alignment between both types of evaluation. In fig. 6, we see from the negative correlation between mAP and NMAC frequency that the model is producing downstream results that are consistent at a high level with the test set results. As a specific example, across all times of day, both models resulted in the highest NMAC rate and lowest mAP on late afternoon samples. However, the downstream task evaluation results in other cases that do not line up with the test set evaluation, demonstrating the need for both types of evaluation to truly capture the performance of the model. Even though the models produced higher mAP on Boeing 737-800 images

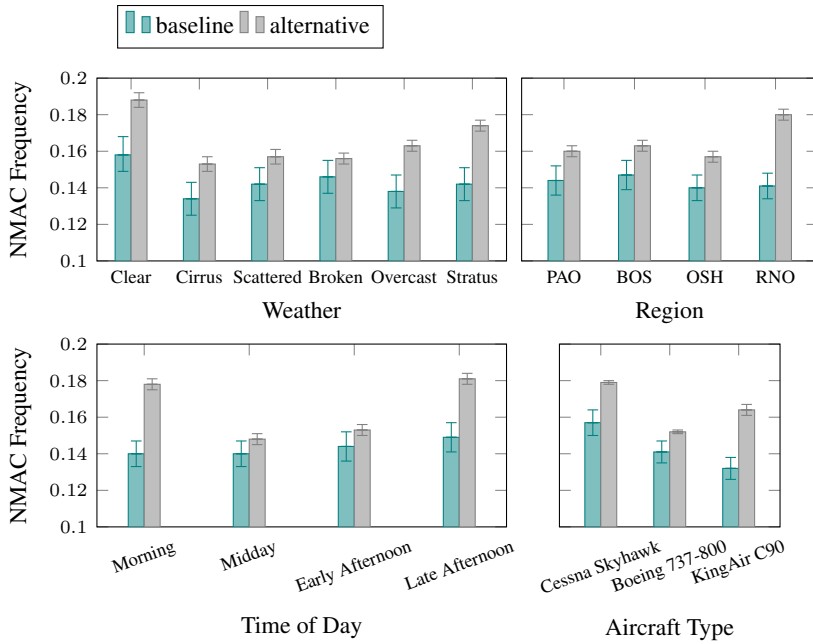

Figure 5: Rate of NMAC for the baseline and alternative models. Error bars represent standard error.

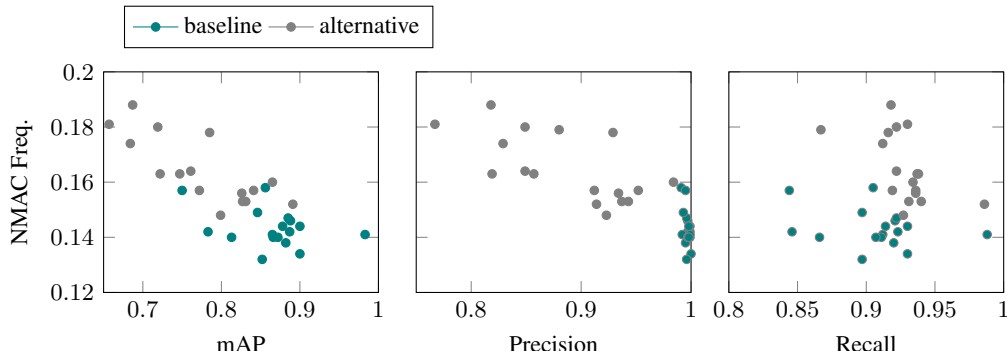

Figure 6: Correlation between downstream task evaluation and test set evaluation metrics.

than the other aircraft types, the NMAC frequency for Boeing 737-800 fell between the frequencies for the other aircraft types. The lack of precise predictability of the models' performance on the downstream task based on the test set evaluation reinforces the value of comprehensive evaluation for real-world high-stakes models.

## 5  Conclusion

In this work, we outlined the AVOIDDS dataset, our associated vision-based aircraft detection models, test set evaluation, and closed-loop simulation. Via evaluation of our baseline model on the test set and downstream task and comparison to the alternative model, we validated the need for sophisticated training and evaluation functionalities, particularly when it comes to real-world, high risk applications in which the margin for error is minimal. These components, while centered around aviation as an application, serve as an example of an accessible, functioning benchmark for designing and refining machine learning-based perception systems using both standard and task-specific evaluation metrics. The main potential negative impact of our work is premature deployment of models that have not been thoroughly trained and tested, resulting in unsafe conditions for the passengers or other aircraft. While we make an effort to cover as many environmental conditions as possible, we cannot guarantee that AVOIDDS covers all conditions aircraft will experience when deployed in the real world. We

also use simulated images, and while X-Plane 11 is photorealistic, the sim-to-real gap should be further investigated. Future work could also explore accounting for state uncertainty in the controller to improve overall performance. We hope AVOIDDS motivates further work on comprehensive benchmarks that can directly impact the training and deployment of complex models.

## Acknowledgments and Disclosure of Funding

The NASA University Leadership Initiative (grant #80NSSC20M0163) provided funds to assist the authors with their research. This research was also supported by the National Science Foundation Graduate Research Fellowship under Grant No. DGE–1656518. Any opinion, findings, and conclusions or recommendations expressed in this material are those of the authors and do not necessarily reflect the views of any NASA entity or the National Science Foundation.

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

## A  Encounter Model Details

Table 1: Parameters for data generation.

| Parameter | Minimum | Maximum | Unit |
|---|---|---|---|
| Ownship Horizontal Speed | 60 | 70 | m/s |
| Intruder Horizontal Speed | 60 | 70 | m/s |
| Horizontal Miss Distance | 0 | 100 | meters |
| Vertical Miss Distance | $-30$ | 30 | meters |
| Relative Heading | 100 | 260 | degrees |

In order to simplify the representation of encounters in this study, we adopt a model where the ownship and intruder aircraft move along straight-line trajectories with constant horizontal speeds. To generate an encounter, we follow a two-step process. First, we randomly sample a set of encounter features from uniform distributions within the specified ranges presented in table 1. We then use these features to generate trajectories for both the ownship and intruder aircraft. The horizontal and vertical miss distance parameters indicate the range between the ownship and intruder aircraft and their relative altitude at the point of closest approach. These distances are deliberately chosen to ensure that all encounters result in a near mid-air collision (NMAC) if no collision avoidance action is taken.

Each simulated encounter has a duration of $50$ seconds, with the closest point of approach occurring $40$ seconds into the encounter. The range of relative headings is selected to generate encounters that are nearly head-on, with the intruder aircraft typically within the ownship's field of view. The features specified in table 1 completely determine the relative trajectories of the ownship and intruder aircraft. Once the relative trajectories are generated, we randomly position both trajectories around the origin region by applying rotations and shifts.

## B  Dataset Details

The AVOIDDS dataset was generated to include 72,000 images, of which 64,800 are training images and 7200 are validation images. These images are accompanied by 72,000 label files. Table 2 shows the number of images for each variable category while table 3 shows the exact distributions used to generate the images. Equal amounts of each cloud covering, region, and aircraft type are represented, and the other variables were randomized for each image.

The range between the ownship and intruder aircraft was sampled from a gamma distribution with shape and scale parameters dependent on the aircraft type. We sampled the distances for the Cessna Skyhawk and King Air C90 from a gamma distribution with shape 2 and scale 200, and the Boeing 737-800 distances were sampled from a gamma distribution with shape 3 and scale 200. The expected value of $\Gamma(3, 200)$ is about $200\,\mathrm{m}$ more than that of $\Gamma(2, 200)$ which we intended to account for the larger size of the Boeing 737-800 aircraft relative to the smaller aircraft. A gamma distribution allows us to sample from a distribution that is skewed toward closer ranges, where aircraft are more likely to be visible in the image. To ensure that the aircraft were not positioned too close together, we verified that the sampled values were greater than $20\,\mathrm{m}$ for the Cessna Skyhawk and King Air C90 and $50\,\mathrm{m}$ for the Boeing 737-800. The position vertically and horizontally of the intruder was sampled uniformly within the ownship field of view. The time of day for each sample was randomized between 08:00 and 17:00 on January 1st in each respective location's local time. We split the day into $4$ time windows for evaluation purposes: morning (08:00-10:00), midday (10:00-13:00), afternoon (13:00-15:00), and late afternoon (15:00-17:00).

## C  Additional Results

Table 4 shows the specific test set evaluation values for the baseline and alternative models discussed, specifically precision, recall, and mAP. These can be compared to the downstream task metrics shown in table 5, which includes the frequencies of NMAC and advisory.

Table 2: AVOIDDS dataset overview.

| Attribute | Value | Number of images | | |
| | | Total | Training | Validation |
|---|---|---|---|---|
| All | - | 72,000 | 64,800 | 7200 |
| Clouds | Clear | 12,000 | 10,800 | 1200 |
| | High Cirrus | 12,000 | 10,800 | 1200 |
| | Scattered | 12,000 | 10,800 | 1200 |
| | Broken | 12,000 | 10,800 | 1200 |
| | Overcast | 12,000 | 10,800 | 1200 |
| | Stratus | 12,000 | 10,800 | 1200 |
| Region | Palo Alto, CA (PAO) | 18,000 | 16,200 | 1800 |
| | Boston, MA (BOS) | 18,000 | 16,200 | 1800 |
| | Oshkosh, WI (OSH) | 18,000 | 16,200 | 1800 |
| | Reno, NV (RNO) | 18,000 | 16,200 | 1800 |
| Aircraft type | Cessna Skyhawk | 24,000 | 21,600 | 2400 |
| | Boeing 737-800 | 24,000 | 21,600 | 2400 |
| | King Air C90 | 24,000 | 21,600 | 2400 |
| Range | $0-150\,\mathrm{m}$ | 9124 | 8268 | 856 |
| | $150-500\,\mathrm{m}$ | 35,932 | 32,303 | 3629 |
| | $>500\,\mathrm{m}$ | 26,944 | 24,229 | 2715 |
| Intruder rel. alt. | Below | 36,048 | 32,482 | 3566 |
| | Above | 35,952 | 32,318 | 3634 |
| Time of day | Morning | 15,930 | 14,385 | 1545 |
| | Midday | 24,142 | 21,722 | 2420 |
| | Afternoon | 15,954 | 14,269 | 1685 |
| | Late Afternoon | 15,974 | 14,424 | 1550 |

Table 3: Parameters for the AVOIDDS Dataset.

| Parameter | Minimum | Maximum | Distribution | Unit |
|---|---|---|---|---|
| Ownship distance east/north from origin | $-5000$ | 5000 | $\mathcal{U}(-5000, 5000)$ | meters |
| Ownship distance vertically from origin | $-1000$ | 1000 | $\mathcal{U}(-1000, 1000)$ | meters |
| Ownship and intruder heading | 0 | 360 | $\mathcal{U}(0, 360)$ | degrees |
| Ownship pitch | $-30$ | 30 | $\mathcal{N}(0, 5)$ | degrees |
| Ownship roll | $-45$ | 45 | $\mathcal{N}(0, 10)$ | degrees |
| Time of day | 08:00 | 17:00 | $\mathcal{U}(08:00, 17:00)$ | hours |

# D    Experiment Reproduction

The experiments in this work can be reproduced using the AVOIDDS repository, available at this link: https://github.com/sisl/VisionBasedAircraftDAA.

## D.1    Test Set Evaluation

Steps for how to reproduce the test set evaluation experiment results are as follows:

1. **Download AVOIDDS dataset:**  Download and extract the AVOIDDS dataset (https://purl.stanford.edu/hj293cv5980) and place the folder in a convenient location.

2. **Download and setup the code repository:**  Download the code repository (https://github.com/sisl/VisionBasedAircraftDAA). Navigate to the `src/model` directory and run `pip3 install -r requirements.txt` to install the necessary dependencies for test set evaluation.

Table 4: Test set evaluation results for baseline and alternative model

| Attribute | Value | Baseline model | | | Alternative model | | |
|---|---|---|---|---|---|---|---|
| | | Precision | Recall | mAP | Precision | Recall | mAP |
| All | - | 0.997 | 0.907 | 0.866 | 0.884 | 0.928 | 0.764 |
| Clouds | Clear | 0.991 | 0.905 | 0.856 | 0.818 | 0.918 | 0.687 |
| | High Cirrus | 1.000 | 0.930 | 0.900 | 0.943 | 0.931 | 0.827 |
| | Scattered | 0.997 | 0.923 | 0.887 | 0.952 | 0.936 | 0.841 |
| | Broken | 0.997 | 0.921 | 0.888 | 0.934 | 0.936 | 0.826 |
| | Overcast | 0.995 | 0.920 | 0.882 | 0.857 | 0.937 | 0.747 |
| | Stratus | 0.999 | 0.846 | 0.783 | 0.829 | 0.912 | 0.684 |
| Region | Palo Alto, CA (PAO) | 0.999 | 0.930 | 0.900 | 0.984 | 0.934 | 0.865 |
| | Boston, MA (BOS) | 0.996 | 0.922 | 0.885 | 0.819 | 0.938 | 0.722 |
| | Oshkosh, WI (OSH) | 0.999 | 0.866 | 0.813 | 0.912 | 0.919 | 0.772 |
| | Reno, NV (RNO) | 0.992 | 0.912 | 0.865 | 0.849 | 0.922 | 0.719 |
| Aircraft Type | Cessna Skyhawk | 0.995 | 0.844 | 0.750 | 0.880 | 0.867 | 0.643 |
| | Boeing 737-800 | 0.999 | 0.988 | 0.983 | 0.914 | 0.986 | 0.891 |
| | King Air C90 | 0.996 | 0.897 | 0.852 | 0.849 | 0.922 | 0.761 |
| Range | $0-150\,\mathrm{m}$ | 0.999 | 0.983 | 0.979 | 0.914 | 0.997 | 0.909 |
| | $150-500\,\mathrm{m}$ | 0.997 | 0.960 | 0.942 | 0.879 | 0.979 | 0.843 |
| | $>500\,\mathrm{m}$ | 0.995 | 0.818 | 0.714 | 0.881 | 0.838 | 0.621 |
| Intruder rel. alt. | Below | 0.995 | 0.844 | 0.847 | 0.905 | 0.920 | 0.764 |
| | Above | 0.998 | 0.917 | 0.884 | 0.863 | 0.937 | 0.763 |
| Time of day | Morning | 0.997 | 0.911 | 0.872 | 0.929 | 0.916 | 0.785 |
| | Midday | 0.998 | 0.907 | 0.866 | 0.923 | 0.927 | 0.799 |
| | Afternoon | 0.998 | 0.914 | 0.878 | 0.937 | 0.940 | 0.831 |
| | Late Afternoon | 0.993 | 0.897 | 0.846 | 0.767 | 0.930 | 0.657 |

3. **Begin evaluation:** Run `python3 eval.py -o baseline_results -d [PATH TO AVOIDDS DATASET]`. Results for baseline model evaluation will appear in `baseline_results.txt`. Run `python3 eval.py -o alternative_results -m "../../models/alternative.pt" -d [PATH TO AVOIDDS DATASET]` to evaluate the alternative model with results outputting to `alternative_results.txt`.

## D.2 Downstream Task Evaluation

Steps for how to reproduce the above downstream task experiment results are as follows:

1. **Download and setup the code repository:** Download the code repository (https://github.com/sisl/VisionBasedAircraftDAA). Navigate to the `src/simulator` directory and run `pip3 install -r requirements.txt` to install the necessary dependencies for downstream task evaluation.

2. **Setup X-Plane and aircraft:** Follow the instructions in the "Setup X-Plane" section of the benchmark repository README file (https://github.com/sisl/VisionBasedAircraftDAA/tree/main/src/simulator), setting the intruder aircraft as the one with which you would like to simulate encounters.

3. **Set variables for simulation:** At the bottom of the `simulate.py` file, uncomment and set the variables in the "BULK SIMULATION VARIABLE SETUP". Set `args.craft` to the aircraft you set in X-Plane in the previous step. Set `args.model_path` to the desired model as well.

4. **Begin simulation:** In the command line, run `./simulate.sh` and toggle your screen such that the X-Plane window is visible and full-screen. You will need to leave this visible for the entirety of the simulation.

5. **Repeat:** Repeat steps 2 through 4 for each desired aircraft and model.

Table 5: Downstream task evaluation results for baseline and alternative model

| Attribute | Value | Total Encs | Baseline model | | Alternative model | |
|---|---|---|---|---|---|---|
| | | | NMAC frq | Advisory frq | NMAC frq | Advisory frq |
| All | - | 8640 | 0.143 | 0.187 | 0.165 | 0.191 |
| Clouds | Clear | 1440 | 0.158 | 0.181 | 0.188 | 0.186 |
| | High Cirrus | 1440 | 0.134 | 0.192 | 0.153 | 0.206 |
| | Scattered | 1440 | 0.142 | 0.191 | 0.157 | 0.206 |
| | Broken | 1440 | 0.146 | 0.184 | 0.156 | 0.194 |
| | Overcast | 1440 | 0.138 | 0.188 | 0.163 | 0.183 |
| | Stratus | 1440 | 0.142 | 0.185 | 0.174 | 0.172 |
| Region | Palo Alto, CA (PAO) | 2160 | 0.144 | 0.199 | 0.160 | 0.203 |
| | Boston, MA (BOS) | 2160 | 0.147 | 0.185 | 0.163 | 0.198 |
| | Oshkosh, WI (OSH) | 2160 | 0.140 | 0.180 | 0.157 | 0.190 |
| | Reno, NV (RNO) | 2160 | 0.141 | 0.183 | 0.180 | 0.173 |
| Aircraft Type | Cessna Skyhawk | 2880 | 0.157 | 0.124 | 0.179 | 0.130 |
| | Boeing 737-800 | 2880 | 0.141 | 0.280 | 0.152 | 0.287 |
| | King Air C90 | 2880 | 0.132 | 0.156 | 0.164 | 0.156 |
| Time of day | Morning | 2160 | 0.140 | 0.186 | 0.178 | 0.193 |
| | Midday | 2160 | 0.140 | 0.188 | 0.148 | 0.195 |
| | Afternoon | 2160 | 0.144 | 0.188 | 0.153 | 0.194 |
| | Late Afternoon | 2160 | 0.149 | 0.185 | 0.181 | 0.182 |

