# OpenReview forum: "AVOIDDS: Aircraft Vision-based Intruder Detection Dataset and Simulator"
_NeurIPS.cc/2023/Track/Datasets_and_Benchmarks — NeurIPS 2023 Datasets and Benchmarks Poster_

### Official Review · Reviewer_hmNh · 2023-07-21
**This submission provides a new labeled dataset (AVOIDDS) for aircraft vision-based intruder detection and avoidance. This dataset is well designed, well documented and includes an important variety of distribution shifts due to changes in environmental conditions (weather, time and location).**

**Rating:** 8
**Confidence:** 3
**Clarity:** The manuscript is well-written and ea…

**Strengths:**

The study is based on a new dataset of 72000 images (AVOIDDS) which includes an important variety of distribution shifts due to changes in environmental conditions (weather, time and location). Each image contains one intruder (different aircraft types are used) which is randomly positioned within the frame. The dataset is very complete and well documented. It is thus likely to be used by researchers in the field of aviation. The authors also provide the code for the customizable generation of additional images.

The use of a simulator gives access to ground-truth position and camera data and thus permits the automatic generation of bounding box labels without manual labeling. The performances of pre-trained models on the AVOIDDS dataset are not only evaluated using classical metrics (e.g., mean average precision) but also in the context of a higher-level task (collision avoidance), which is very important for use in safety-critical conditions.


**Additional Feedback:**

Why did the authors only focus on 4 geographic regions and 3 aircraft types? Is it because these are the only options in the X-Plane 11 flight simulator?

**Correctness:**

The dataset is constructed in a sound way and benefits from the use a publicly available commercial flight simulator. The use of this simulator gives an automatic access to ground-truth position and camera data without any manual labeling. It is however important to further characterize how realistic are the generated images (see my point in the ‘Opportunities for improvements’ section above). In the data generation section (and also in the ‘Dataset details’ section of the supplementary materials), it was not clear to me why the ranges between the ownship and intruder aircraft were sampled from Gamma distributions and how the parameters of these Gamma distributions were chosen. The evaluations of the performances for detections and collision avoidances are performed appropriately.

**Documentation:**

The dataset is publicly available and its generation is based on a photo-realistic flight simulator (X-Plane 11) that is also publicly available. The way the images were created with X-Plane 11 is clearly documented and the authors provide the code to generate new images under various conditions. The structure of the data is also clearly described in the supplementary materials. It is in the YOLO format with subdirectories for images and labels for both the training and validation sets. For each image, metadata provides the position of the aircraft intruder as well as the environmental details.

The training based on YOLOv8 is correctly described. The encounter model and simulator are based on a general interface in order to facilitate the future incorporation of more complex models.

The authors provide a repository to reproduce the experiments described in their submission and detail the different steps to follow in the supplementary materials.


**Ethics:**

The dataset focuses on aircraft intruder images generated from a photo-realistic flight simulator and it does not contain any human information. In my opinion, there is thus no ethical concerns with this submission.

**Limitations:**

The authors report a few limitations in their conclusion. They acknowledge that their simulations might not cover all the environmental conditions an aircraft experiences in real-world conditions. They also come back on the fact that simulated images might still differ from real images. Finally, they propose that future works should explore how performances on the collision avoidance task could be improved by taking into account state uncertainty in the controller. These three points could have been discussed in more details in the manuscript.

**Opportunities For Improvement:**

In my opinion, it would be useful that the authors provide more information regarding the quality of the images generated by the X-Plane 11 flight simulator. In particular, how do they compare to real images captured by aircrafts (which are potentially noisy and/or degraded)? This point is very important to assess the validity of the study in real-world conditions.

Also, the authors used a simple pairwise encounter model where aircrafts follow straight line trajectories. I think it would be interesting that the authors discuss the limits of this model and how it can be improved with more complex models in futures works. More information on the controller and on its limitations could also be useful.


**Relation To Prior Work:**

The authors provide a fair description of previous object detection models which were usually evaluated using isolated metrics (e.g., mean average precision) and less often on higher-level decision-making tasks. They clearly describe the contribution of their dataset which expands on previous applications in the aviation domain to evaluate performances in the specific context of aircraft collision avoidance.

**Summary And Contributions:**

This submission provides a public and labeled dataset (AVOIDDS) for aircraft vision-based intruder detection and avoidance. This dataset consists in numerous images with an intruder aircraft at different positions in the frame and with various lighting conditions, weather conditions, relative geometries and geographic locations. Each image was generated with a photo-realistic flight simulator (X-Plane 11) and is annotated with metadata that describes its environmental conditions. The authors also provide an interface to evaluate the performances of pre-trained models on the dataset and a benchmark based on YOLOv8.  Finally, they propose a close-loop simulator to evaluate trained detection models in the practical case of collision avoidance.

---

> ### Author Response · Authors · 2023-08-21
> **Response to Reviewer hmNh**
>
> We thank the reviewer for their time in reviewing the paper and for the helpful feedback. We provide responses below:
>
> **Question:** In particular, how do they compare to real images captured by aircrafts (which are potentially noisy and/or degraded)?
>
> **Answer:** X-Plane 11 is designed to be a photorealistic simulator, so the images capture a number of real-world effects. However, as noted by the reviewer, some artifacts that may be present in real-world images may be missing. These effects are often dependent on the performance characteristics of the camera and should be considered on a case by case basis. We note this sim-to-real gap in the conclusion (see paper). The photorealistic nature of X-Plane 11 indicates that research findings on the AVOIDDS dataset should also hold on a real-world dataset.
>
> **Question:** the authors used a simple pairwise encounter model where aircrafts follow straight line trajectories. I think it would be interesting that the authors discuss the limits of this model and how it can be improved with more complex models in futures works.
>
> **Answer:** The straight-line encounter model provided in the benchmark simulator is a good starting point for testing vision-based collision avoidance systems. However, as the reviewer pointed out, it has some limitations in that it does not fully capture all aircraft behavior. Some previous works have focused on creating data-driven statistical encounter models that adequately capture aircraft behavior [1]. For example, researchers have developed models of manned aircraft [1] and small unmanned aircraft [2]. These models are publicly available on the Airspace Encounter Models Github (https://github.com/Airspace-Encounter-Models). We have designed our code such that encounters are input to the simulation using a standardized file format which allows users to sample encounters from any encounter model, convert them to the proper format, and simulate them. We modified the end of the Encounter Model paragraph of Section 3.3 to incorporate an abridged version of this discussion (see revised paper).
>
> **Question:** More information on the controller and on its limitations could also be useful.
>
> **Answer:** We provide an interface to use the control policy defined in the VerticalCAS repository, which models the collision avoidance problem as a Markov Decision Process (MDP). As noted in Section 3.3, VerticalCAS was loosely inspired by the ACAS X family of collision avoidance systems, which also rely on MDPs [3-4]. The version of ACAS X created for manned aircraft became an international standard in 2018. Therefore, VerticalCAS is a strong baseline inspired by real-world systems; however, since it is a notional example designed for research purposes, one limitation is that it has not been put through the same rigorous testing and validation as ACAS X. We modified the end of the Controller paragraph of Section 3.3 to incorporate an abridged version of this discussion (see revised paper).
>
> **Question:** In the data generation section (and also in the ‘Dataset details’ section of the supplementary materials), it was not clear to me why the ranges between the ownship and intruder aircraft were sampled from Gamma distributions and how the parameters of these Gamma distributions were chosen.
>
> **Answer:** We thank the reviewer for identifying this point of confusion. We describe the reasoning behind the selection of these distributions in Appendix B. We have also included some additional text to further clarify why we selected gamma distributions (see revised Appendix).
>
> **Question:** Why did the authors only focus on 4 geographic regions and 3 aircraft types? Is it because these are the only options in the X-Plane 11 flight simulator?
>
> **Answer:** X-Plane 11 has scenery for regions covering much of the world and a few dozen aircraft types. We deliberately selected a subset of these regions and aircraft types to provide a diverse set of objects and backgrounds. If desired, our data generation code could be used with X-Plane 11 to generate new datasets in different regions or using different aircraft types.
>
> **References**
>
> [1] M. J. Kochenderfer, M. W. M. Edwards, L. P. Espindle, J. K. Kuchar, and J. D. Griffith, “Airspace Encounter Models for Estimating Collision Risk,” Journal of Guidance, Control, and Dynamics, vol. 33, no. 2, pp. 487–499, Apr. 2010.
>
> [2] A. Weinert and N. Underhill, “Generating Representative Small UAS Trajectories using Open Source Data,” in Digital Avionics Systems Conference (DASC), 2018, pp. 1–10.
>
> [3] M. P. Owen, A. Panken, R. Moss, L. Alvarez, and C. Leeper, “ACAS Xu: Integrated collision avoidance and detect and avoid capability for UAS,” in Digital Avionics Systems Conference (DASC), IEEE, 2019, pp. 1–10.
>
> [4] L. E. Alvarez, I. Jessen, M. P. Owen, J. Silbermann, and P. Wood, “ACAS sXu: Robust decentralized detect and avoid for small unmanned aircraft systems,” in Digital Avionics Systems Conference (DASC), IEEE, 2019, pp. 1–9.

---

> > ### Comment · Reviewer_hmNh · 2023-08-29
> >
> > I thank the authors for their clarifications. They improve the study but I feel that the connexion with real images is still difficult to make. I also agree with reviewer s82Z that a video dataset would have been more useful. For this reason, I retain my ratings.

---

### Official Review · Reviewer_s82Z · 2023-08-01
**Flight intruder detection Is Important, but we prefer video-based methods**

**Rating:** 5
**Confidence:** 4
**Correctness:** Most of the presentations are correct.
**Clarity:** Clear.

**Strengths:**

(1) The task is clear and important for civil aviation;
(2) The dataset contains object in different backgrounds;
(3) The baseline is quite popular and widely acknowledged.

**Additional Feedback:**

This task is important. The authors are encouraged to expand the current version to video data.

**Documentation:**

Yes.

**Ethics:**

No.

**Limitations:**

Not really. Video data is more effective for this task. In addtion, there are also ad hoc flying object detection methods that are more accurate than the YOLOv8 in this task.

**Opportunities For Improvement:**

(1) Realted work is far from adequate. There are also tiny object detection methods, and uav2uav methods that should be reviewed.
(2) Why images? Videos is more practical and effective.
(3) Though many, the image samples are not adequate enough for practical applications (The task can hardly bare mistakes).
(4) Any other methods beyond YOLOv8?
(5) The challenges and main problems are expected to be highlighted and discussed.

**Relation To Prior Work:**

Not yet. The related work part should be enriched. Many flying object detection methods should be reviewed.

**Summary And Contributions:**

The work presents the AVOIDDS dataset: associated vision-based aircraft detection and simulator. The task is clear to understand, which requires flying object detection in the images. The dataset has collected many object images in different scenarios.

---

> ### Author Response · Authors · 2023-08-21
> **Response to Reviewer s82Z**
>
> We thank the reviewer for their time in reviewing the paper and for the helpful feedback. We provide responses below:
>
> **Response to suggestions for related work:** We thank the reviewer for pointing us to the suggested literature on UAV to UAV and tiny object tracking [1-3]. We see the following primary connections between this line of work and our present work:
>
> 1. These works focus on similar tasks involving aerial imagery. Some focus on tracking generic objects in aerial imagery [1][2], while the UAV to UAV work focuses specifically on tracking other UAVs [3].
> 2. As the reviewer pointed out, these works focus on tracking objects using sequences of images (i.e. videos), while our benchmark dataset consists of uncorrelated image frames. In contrast, when we simulate encounters using our benchmark simulator, we produce sequences of time-correlated images (videos) that closely mimic the types of scenes included in the UAV to UAV tracking benchmark [3].
> 3. While the UAV to UAV work has the advantage of using images taken in the real world, producing our benchmark images using the photorealistic X-Plane 11 flight simulator has other advantages. First, since the simulator provides access to the ground truth location of the aircraft in the frame, we can automatically compute bounding boxes without needing manual labeling. The simulator also allows us to customize environmental conditions, which is a major feature of AVOIDDS. Furthermore, the video frames provided in the UAV to UAV benchmark do not allow for simulation of counterfactuals or new encounter scenarios, which can be done using AVOIDDS.
>
> We have integrated these references and an abridged version of this discussion into the Vision-based aircraft detect-and-avoid paragraph of Section 2 (see revised paper).
>
> **Question:** Why images? Videos is more practical and effective.
>
> **Answer:** A number of recent works in machine learning-based object detection operate on single image frames [4-6]. By supplying a dataset of single image frames, we encourage the participation of the broader computer vision community. As noted by the reviewer, operating on videos allows for tracking of objects over time. In the single frame approaches, we could account for the temporal component by quantifying uncertainty in the predictions, assuming a dynamics model, and implementing filtering techniques. In addition, producing video data requires an assumption of the flight path of each aircraft. We include this assumption by specifying an encounter model and simulating aircraft trajectories in the closed-loop simulator. These simulations produce sequences of images, which could be saved as video data and used to train or test vision-based UAV tracking methods. We thank the reviewer for pointing out this area of related work to elaborate on, and we hope that the discussion we have added to related work provides helpful additional context.
>
> **Question:** Though many, the image samples are not adequate enough for practical applications (The task can hardly bare mistakes).
>
> **Answer:** We agree with the reviewer that the 72,000 images present in the AVOIDDS dataset are not adequate to ensure the level of safety required to confidently deploy a system in aerospace applications. However, we also provide data generation and encounter simulation code that can be used to generate an arbitrary number of datapoints and scenarios. These features allow for safety evaluation of an already very safe system. The AVOIDDS dataset is intended as a starting point to identify challenges and explore new ideas in the research community. We also note in the conclusion the risk of premature deployment without thorough testing (see paper).s
>
> **Question:** Any other methods beyond YOLOv8?
>
> **Answer:** While there are a number of other object detection methods [4-6], the primary focus of our work is the dataset and encounter simulator. We chose YOLOv8 as a state-of-the-art example to demonstrate these components. However, to account for this limit in scope, we designed the dataset and simulator so that other models could easily be plugged in for training and evaluation.
>
> **References**
>
> [1] Mueller, Matthias, et al. "A benchmark and simulator for UAV tracking." in European Conference on Computer Vision, Springer, 2016.
>
> [2] Li, Siyi, et al. "Visual object tracking for unmanned aerial vehicles: A benchmark and new motion models." AAAI Conference on Artificial Intelligence, 2017.
>
> [3] Wang, Yong, et al. "A UAV to UAV tracking benchmark." in Knowledge-Based Systems, 261 (2023): 110197.
>
> [4] L. Liu, et al. “Deep learning for generic object detection: A survey,” International Journal of Computer Vision, vol. 128, no. 2, pp. 261–318, 2020.
>
> [5] R. Girshick, et al. “Rich feature hierarchies for accurate object detection and semantic segmentation,” in Computer Vision and Pattern Recognition, 2014.
>
> [6] W. Liu, et al. “SSD: Single shot multibox detector,” in European Conference on Computer Vision, Springer, 2016.

---

> > ### Comment · Reviewer_s82Z · 2023-08-25
> >
> > Thanks to the authors for the explanations. I am glad to see the improvements. Nevertheless, for this specific application, video is much more suitable than images. Although static image object detection is still important for general object detection, but I can hardly image that a drone would prefer to use static image to observe the real-time real-world. I am sorry, but I am afraid that I have to retain my rating.
> > (I strongly hope that the authors can make the dataset into a video one. That would be very useful, and would benefit the related fields. Thanks!)

---

### Decision · Program_Chairs · 2023-09-22

**Decision:**

Accept (Poster)

**Comment:**

The paper presents a well-written and accessible proposal for a simulator and the AVOIDSS dataset, offering valuable resources for training and evaluating high-risk aircraft detection systems; however, more through benchmarking and comparative analysis with other datasets, and the potential for an extended video dataset should be explored to enhance its utility. Given the accessibility of the platform future users will be able to generate their choice of dataset.